# Impact of the Treatment of *Serenoa repens, Solanum lycopersicum, Lycopene* and *Bromelain* in Combination with Alfuzosin for Benign Prostatic Hyperplasia. Results from a Match-Paired Comparison Analysis

Luca Lambertini [1,2], Fabrizio Di Maida [1,2], Riccardo Tellini [1,2], Claudio Bisegna [1,2], Francesca Valastro [1,2], Antonio Andrea Grosso [1,2], Sabino Scelzi [1,2], Francesco Del Giudice [3], Matteo Ferro [4], Giacomo Maria Pirola [5], Marilena Gubbiotti [5], Lorenzo Masieri [1,2], Gian Maria Busetto [3], Ottavio de Cobelli [4], Andrea Minervini [1,2] and Andrea Mari [1,2,*]





[1] Unit of Urological Oncologic Minimally-Invasive Robotic Surgery and Andrology, Careggi Hospital, 50134 Florence, Italy; l.lambertini7@gmail.com (L.L.); fabridima90@gmail.com (F.D.M.); riccatello@gmail.com (R.T.); claudio.bisegna@unifi.it (C.B.); francesca.valastro@unifi.it (F.V.); grossoantonioandrea@gmail.com (A.A.G.); sabino.scelzi@unifi.it (S.S.); lorenzo.masieri@unifi.it (L.M.); andreamine@libero.it (A.M.)

[2] Department of Experimental and Clinical Medicine, University of Florence, 50134 Florence, Italy

[3] Department of Urology, Sapienza University of Rome, 00185 Rome, Italy; francesco.delgiudice@uniroma1.it (F.D.G.); gianmariabusetto@gmail.com (G.M.B.)

[4] Division of Urology, European Institute of Oncology, IRCCS, 20133 Milan, Italy; matteo.ferro@ieo.it (M.F.); ottaviodecobelli@gmail.com (O.d.C.)

[5] Department of Urology, San Donato Hospital, 52100 Arezzo, Italy; gmo.pirola@gmail.com (G.M.P.); marilena.gubbiotti@gmail.com (M.G.)

* Correspondence: andreamari08@gmail.com; Tel.: +39-0552758011 or +39-3926725196; Fax: +39-0552758014

**Abstract:** Background: Phytotherapeutic agents aroused an increasing interest either as alternative or in addition to conventional therapy in the management of BPH. The aim of the article was to evaluate the clinical and functional changes after add-on treatment with *Serenoa repens* associated with *Solanum lycopersicum, lycopene* and *bromelain* in patients with BPH presenting with mild to moderate LUTS and previously treated only with Alfuzosin over a 6–12 months period. Materials and methods: Between January and July 2019, patients with symptomatic BPH already on treatment with Alfuzosin (Al) 10 mg for at least 6–12-month were enrolled at three academic referral centres, included in a prospective treatment group, and managed with a combination treatment of Al and 6-month daily oral single-tablet supplementation of *Serenoa repens* and *Solanum lycopersicum + lycopene + bromelain* (SeR + SL + Ly + Br). A retrospective control group with comparable baseline characteristics was obtained by performing a propensity score matching from a database of 434 patients managed with Alfuzosin 10 mg/day only over a 6–12 months period between March 2015 and December 2018. IPSS, QoL questionnaires, voiding diary assessment, postvoid residual volume (PVR), maximal (Qmax) and average (Qave) urinary flow rates were evaluated at baseline in the treatment group at the moment of patient accrual, in the control group after 6-month of treatment with alfuzosin, and thereafter at 3 and 6 months in both groups. Results: Overall, 250 patients entered the study (*n* = 125 treatment group; *n* = 125 control group). Total IPSS score significantly decreased at 6-month assessment from baseline in the treatment vs control group (17 [IQR: 12–20] vs 12 [IQR: 9–14], *p* = 0.02) with a significative storage symptoms improvement detected both at 3- (*p* = 0.03) and 6-month evaluation (*p* = 0.001). PVR significantly improved at each follow-up visit with the most relevant reduction at 6-month assessment (125 cc vs. 102 cc; *p* = 0.02). Moreover, a significative improvement in LUTS-related quality of life (QoL) was recorded at 3- and 6-month assessment with a median decrease of −1 and −2 (*p* = 0.05 and *p* = 0.001 respectively) from baseline. Conclusions: Combination treatment with AB and SeR + SL + Ly + Br led to meaningful improvements in LUTS severity compared to AB as monotherapy, after a 6-month treatment period in men with mild to moderate LUTS/BPH.

**Keywords:** benign prostate hyperplasia; inflammation; phytotherapy; *Serenoa repens*; *Solanum lycopersicum*; *lycopene*; *bromelain*; α1 blockers

## 1. Introduction

Benign prostatic hyperplasia (BPH) is one of the most common progressive disorders affecting nearly half of middle-age men and it is frequently associated with lower urinary tract symptoms (LUTS) and patients' quality-of-life (QoL) impairment [1,2].

Although hormonal and vascular age-related changes seem to be involved in disease development, the exact pathophysiology of BPH is still yet to be identified [3]. In this regard, a growing body of evidence showed that the chronic inflammatory cascade activation and the consequent local growth factor production might play a pivotal role in BPH progression [4]. Currently, in agreement with the current international guidelines, alpha1-adrenoceptor blockers (AB), 5-alpha reductase inhibitors (5ARIs) and phosphodiesterase type 5 inhibitor (PDE5I) are extensively used as first-line treatment in BPH patients [5]. However, these pharmacologic approaches are not devoid of adverse effects such as blood pressure modifications and sexuality disorders [6].

In this scenario, an increasing interest on the role of phytotherapeutic and nutraceutical agents either as alternative or in addition to conventional therapy for the management of BPH has evolved during the last decades [7–9]. Particularly, if on the one hand *Serenoa repens* (SeR) represents nowadays one of the most promising extracts [10], on the other hand several recent clinical trials assessed its effectiveness and safety as first-line approach as monotherapy or in combination with other nutraceutical elements such as selenium (Se) and *lycopene* (Ly) [11–13]. Nevertheless, the different extraction techniques and concentration of active ingredients, the absence of long-term follow-up assessment and the lack of a standardized symptoms evaluation could have substantially impaired trustworthiness of the above-mentioned studies, ultimately undermining the final reliability of reported results [14,15].

To address this unmet need, the aim of this multicenter study was to evaluate the clinical and functional changes after add-on treatment with *Serenoa repens* associated with *Solanum lycopersicum*, *lycopene* and *bromelain* in patients with BPH presenting with mild to moderate LUTS and already on treatment with Alfuzosin 10 mg for at least 6–12-month.

## 2. Materials and Methods

### 2.1. Statement of Human Rights

The present study has been approved by the institutional research ethics committee before experiment was started and has been conducted in accordance with the principles set forth in the Helsinki Declaration. The article received the Institutional Review Board approval.

### 2.2. Patients Selection

We prospectively collected and retrospectively analyzed data of consecutive patients affected by BPH presenting with mild to moderate LUTS between January 2019 and July 2019 at three academic referral centres composing the prospective treatment group.

Main inclusion criteria were: (1) age between 40 and 70 years, (2) previous therapy with Alfuzosin 10 mg over a 6–12-month period, (3) international prostate symptom score (IPSS) < 20 with a positive answer to urgency and frequency questions. Exclusion criteria included prostate volume > 100 mL, previous pharmacological or surgical treatment for BPH, histologically confirmed diagnosis of prostate cancer (PCa) and the presence of neurological comorbidities.

A retrospective control group was obtained by performing a propensity score–matched analysis on 434 patients previously managed with Alfuzosin 10 mg/day during a 6–12 months between March 2015 and December 2018. Propensity score was calculated by multivariate

logistic regression model for each case based on the following characteristics evaluated after 6 months of treatment with alfuzosin: age, N° of daytime micturition, prostate volume (cc), Qmax at free uroflowmetry, total PSA, total IPSS score and voiding IPSS subscore.

### 2.3. Study Design and Clinical Outcomes

Each patient in the prospective treatment group signed an informed consent form at baseline after adequate and complete information. All patients were treated with a combination therapy of Alfuzosin 10 mg and 6-month daily oral single-tablet supplementation of *Serenoa repens* and *Solanum lycopersicum* and *lycopene* and *bromelain* (SeR + SL + Ly + Br). One tablet consisted of 400 mg of supercritical $CO_2$ lipidic extract containing 92% of fatty acids sterols, *Solanum lycopersicum* (55 mcg), *lycopene* (10 mg) and *bromelain* (200 mg) (Perlaprost®).

In the retrospective control group, patients' characteristics were analyzed every 3 months during the first 6 months of treatment with Alfuzosin 10 mg/day. IPSS and QoL questionnaires and voiding diary were administered at baseline and after 3 and 6 months from patient accrual. Patients reporting questionnaires and clinical features including prostate volume, serum PSA, postvoid residual volume (PVR), maximal (Qmax) and average (Qave) urinary flow rates were evaluated at baseline and later at 3–6 month follow up assessment (Figure 1).

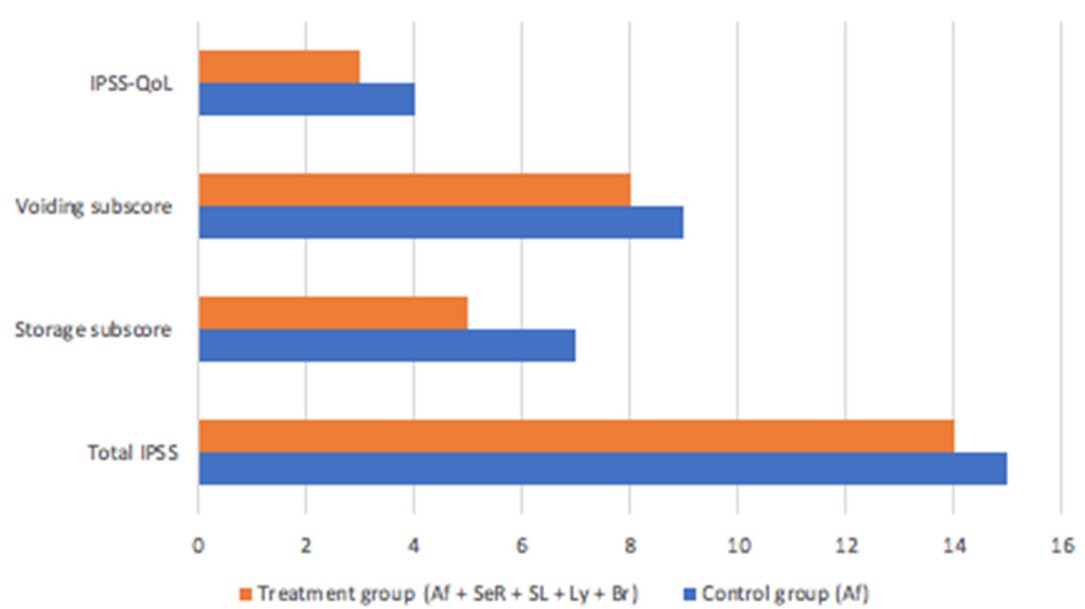

**Figure 1.** International prostate symptoms score (IPSS) and subscores at 3-month follow-up assessments.

The primary endpoint of the study was to evaluate the changes of total IPSS and QoL scores between treatment and control groups at baseline and after 3 and 6 months. A particular attention was paid to changes in storage symptoms. The secondary endpoints were to assess the changes in PVR, total PSA values, voiding diary and median Qmax. The study design is represented within Figure 2.

### 2.4. Statistical Analysis

Collected data were reported as number and percentages or median and interquartile range (IQR). The outcome variables were tested as change from baseline to each visit. Basing on their distribution continuous variables were compared using the Student's t test or the Mann–Whitney U test, as appropriate. Categorical variables were tested with the

$\chi^2$ test. A significance level of $p < 0.05$ was set for all tests. A propensity-score matching was performed to adjust retrospective cohort for baseline variables: age, N° of daytime micturictions, prostate volume (cc), Qmax at non-invasive free uroflowmetry, total PSA, total IPSS score and voiding IPSS subscore. The matching was carried out with a 1:1 ratio with a C statistic of 0.71. Statistical analyses were performed using SPSS version 24 (IBM SPSS Statistics for Macintosh; IBM Corp., Armonk, NY, USA).

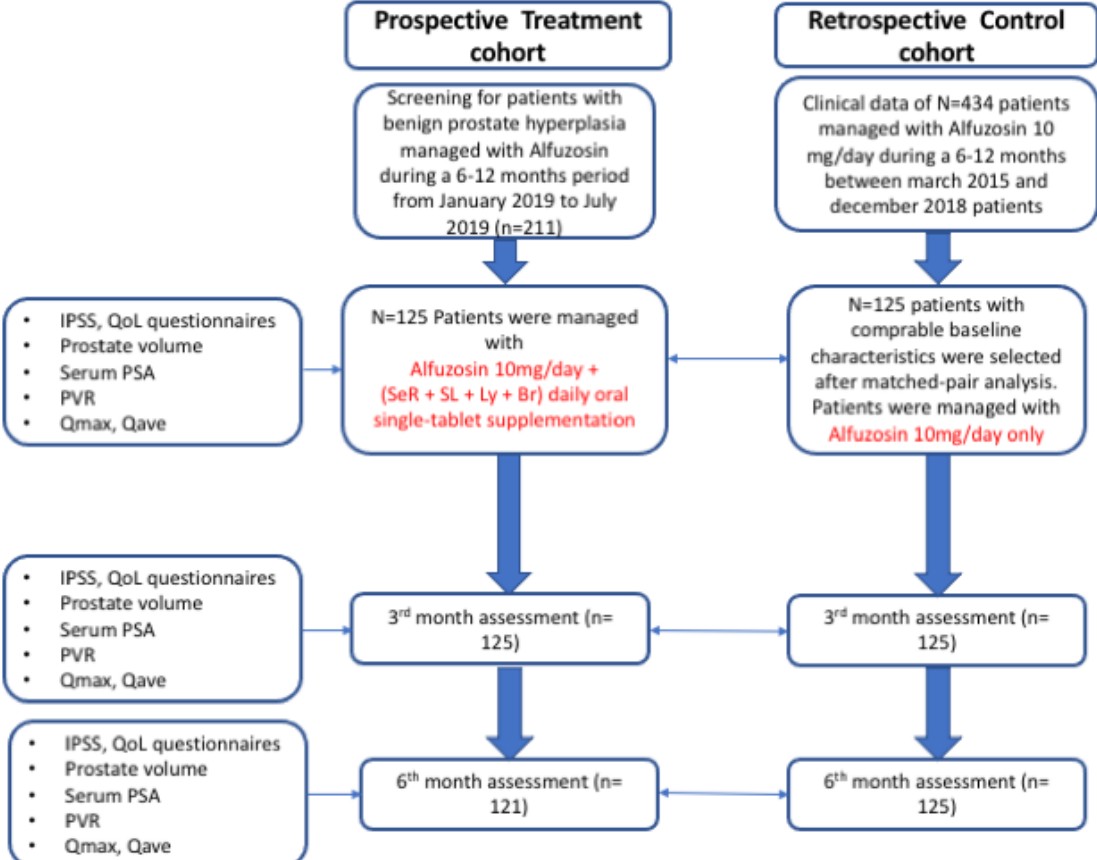

**Figure 2.** Study design.

## 3. Results

Overall, 250 patients entered the study, 125 in the treatment prospective group and 125 in the matched-paired retrospective control group. Median age was 61 (IQR 52–69) years in the treatment group ad 63 (IQR 53–70) years in the control group ($p$-value = 0.57). The groups did not show any significant differences in demographic or baseline clinical parameters (Table 1).

Total IPSS score was found significantly improved at 6-month assessment (12 [IQR: 9–14] vs. 15 [IQR: 12–20], $p = 0.02$) in the treatment group, while no significant differences were found at the third month assessment ($p = 0.07$), (Table 2). Nevertheless, dividing total IPSS into storage and voiding subscores, a significative storage symptoms improvement was detected in the treatment group at 3-month (5 [IQR 4–9] vs. 7 [IQR 5–9]; $p = 0.03$) and 6-month evaluation (4 [IQR 2–7] vs. 7 [IQR 4–9]; $p = 0.001$). Similarly, a significant IPSS-QoL score improvement was noted at 3- and 6-month evaluation in the treatment group with a median decrease of −1 and −2 ($p = 0.05$ and $p = 0.001$ respectively) as compared to the control group (Figure 3). Conversely, no significative differences between the two groups emerged at the 3rd month voiding subscore assessment ($p = 0.06$) while a statistically significant improvement at 6-month evaluation (6 [IQR: 3–9] vs. 9 [IQR: 6–11], $p = 0.03$) in the treatment group was recorded. Table 3 reports median IPSS score and subscores. Median number of night-time micturition significantly decreased equally at 3- and 6-month

assessment (1 [IQR: 1–2], 2 [IQR: 1–4] vs. 3 [IQR: 1–4], *p* = 0.03) in the treatment group, while number of daytime micturition was significantly different between the two groups solely at 6-month evaluation (6 [IQR: 5–9] vs. 8 [IQR: 6–9], *p* = 0.04).

A significative improvement in Qmax (16.7 mL/s [IQR: 13–19] vs. 11.3 mL/s [IQR: 10–15], *p* = 0.001) and Qave (14.2 mL/s [IQR: 10–16] vs. 12 mL/s [IQR 10–14], *p* = 0.05) were found at 6-month evaluation in the treatment group as compared to the control group. In terms of PVR, significant differences were detected at each follow up visit with the most relevant reduction of median PVR at 6-month assessment (102 cc [IQR: 91–116] vs. 127 cc [IQR: 98–139], *p* = 0.02), (Figure 4). Conversely, no significant differences in prostate volume were found during follow-up. As regards total PSA serum levels, the treatment group showed a significant decrease of median values at 3-month evaluation (2.9 ng/mL [IQR 1.9–3.8] vs. 3.4 ng/mL [IQR: 2.2–4.2]; *p* = 0.04) In terms of adverse effects, no significant differences were found as outlined in Table 4.

**Table 1.** Baseline clinical features of the 250 patients completing the study protocol.

| Clinical Features | Treatment Group (Af + SeR + SL + Ly + Br) (*n* = 125) | Control Group (Af) (*n* = 125) | *p*-Value |
|---|---|---|---|
| Age (median, IQR) | 61 (52–69) | 63 (53–70) | 0.57 |
| N° of daytime micturition (median, IQR) | 8 (6–9) | 8 (6–10) | 0.37 |
| N° of night-time micturition (median, IQR) | 3 (1–4) | 2 (1–4) | 0.77 |
| Prostate volume (cc) (median, IQR) | 51 (33–68) | 52 (35–69) | 0.39 |
| Qmax (mL/s) (median, IQR) | 11.3 (10–15) | 12.5 (11–14) | 0.48 |
| Qave (mL/s) (median, IQR) | 11 (10–12) | 10 (9–12) | 0.36 |
| PVR (cc) (median, IQR) | 127 (98–139) | 125 (96–143) | 0.68 |
| Total PSA (ng/mL) (median, IQR) | 3.4 (2.2–4.2) | 3.6 (2.5–4.6) | 0.52 |
| Total IPSS (median, IQR) | 17 (14–19) | 15 (13–17) | 0.74 |
| Storage subscore (median, IQR) | 7(4–9) | 7 (5–9) | 0.85 |
| Voiding subscore (median, IQR) | 9 (6–11) | 10 (8–13) | 0.72 |
| IPSS-QoL (median, IQR) | 4 (3–5) | 4 (4–6) | 0.88 |

**Table 2.** International prostate symptoms score (IPSS) and subscores and respective descriptive statistics at 3-month ([a]) and 6-month ([b]) follow-up assessments.

| Patients Reporting Questionnaires [a] | Control Group (Af) (*n* = 125) | Treatment Group (Af + SeR + SL + Ly + Br) (*n* = 125) | *p*-Value |
|---|---|---|---|
| Total IPSS (median, IQR) | 15 (11–18) | 14 (10–16) | 0.07 |
| Storage subscore (median, IQR) | 7 (5–9) | 5(4–9) | **0.03** |
| Voiding subscore (median, IQR) | 9 (7–11) | 8 (6–9) | 0.06 |
| IPSS-QoL (median, IQR) | 4 (4–6) | 3 (2–4) | **0.05** |
| **Patients Reporting Questionnaires [b]** | **Control Group (Af) (*n* = 125)** | **Treatment Group (Af + SeR + SL + Ly + Br) (*n* = 125)** | ***p*-Value** |
| Total IPSS (median, IQR) | 15 (12–20) | 12 (9–14) | **0.02** |
| Storage subscore (median, IQR) | 7 (4–9) | 4 (2–7) | **0.001** |
| Voiding subscore (median, IQR) | 8 (6–10) | 6 (3–9) | **0.03** |
| IPSS-QoL (median, IQR) | 4 (4–6) | 2 (1–4) | **0.001** |

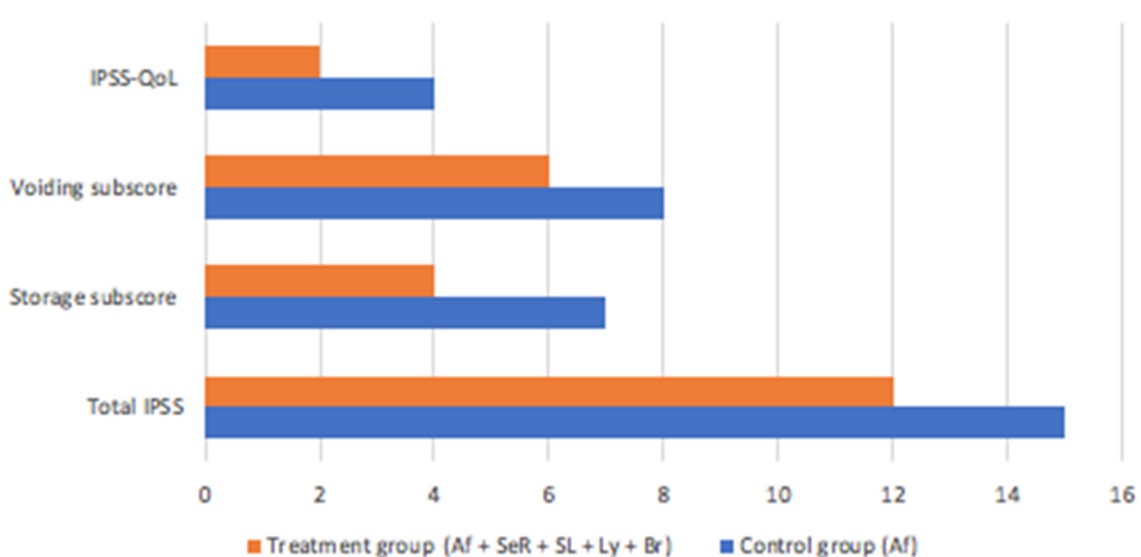

**Figure 3.** International prostate symptoms score (IPSS) and subscores at 6-month follow-up assessments.

**Table 3.** Descriptive statistics of the efficacy parameters at the follow-up assessment at 3 months (ᵃ) and 6 months (ᵇ).

| Clinical Features [a] | Control Group (Af) (*n* = 125) | Treatment Group (Af + SeR + SL + Ly + Br) (*n* = 125) | *p*-Value |
|---|---|---|---|
| N° of daytime micturition (median, IQR) | 8 (6–11) | 7 (6–9) | 0.07 |
| N° of nighttime micturition (median, IQR) | 2 (1–4) | 1 (1–2) | **0.03** |
| Prostate volume (cc) (median, IQR) | 52 (42–64) | 50 (43–63) | 0.39 |
| Qmax (mL/s) (median, IQR) | 13.4 (11–15) | 14.9 (12–18) | 0.08 |
| Qave (mL/s) (median, IQR) | 10 (8–12) | 11.5 (10–14) | 0.09 |
| PVR (cc) (median, IQR) | 119 (96–143) | 112 (75–140) | **0.05** |
| PSA (ng/mL) (median, IQR) | 3.6 (2.5–4.6) | 3.1 (1.8–4.2) | 0.07 |

| Clinical Features [b] | Control Group (Af) (*n* = 125) | Treatment Group (Af + SeR + SL + Ly + Br) (*n* = 125) | *p*-Value |
|---|---|---|---|
| N° of daytime micturition (median, IQR) | 8 (6–11) | 6 (5–9) | **0.04** |
| N° of nighttime micturition (median, IQR) | 2 (1–4) | 1 (1–2) | **0.03** |
| Prostate volume (cc) (median, IQR) | 52 (40–62) | 51 (39–62) | 0.41 |
| Qmax (mL/s) (median, IQR) | 13.8 (12–16) | 16.7 (13–19) | **0.001** |
| Qave (mL/s) (median, IQR) | 12 (10–14) | 14.2 (10–16) | **0.05** |
| PVR (cc) (median, IQR) | 109 (99–123) | 102 (91–116) | **0.02** |
| PSA (ng/mL) (median, IQR) | 3.6 (2.5–4.6) | 2.9 (1.9–3.8) | **0.04** |

**Table 4.** Summary of adverse events within the study population at 6-month assessment.

| Adverse Events | Treatment Group (Af + SeR + SL + Ly + Br) * (*n* = 125) | Control Group (Af) (*n* = 125) | *p* Value |
|---|---|---|---|
| Total, *n* (%) | 16 (13%) | 15 (12%) | 0.84 |
| Ejaculatory disorders | 6 (4.9%) | 6 (4.8%) | 0.92 |
| Postural hypotension | 3 (2.4%) | 4 (3.2%) | 0.78 |
| Headache | 4 (3.3%) | 2 (1.6%) | 0.59 |
| Stomach upset | 3 (2.4%) | 1 (0.8%) | 0.31 |
| Dry mouth | 1 (0.8%) | 2 (1.6%) | 0.42 |

* Af—Alfuzosin, SeR—*Serenoa repens*, SL—*Solanum lycopersicum*, Ly—*lycopene*, Br—*bromelain*.

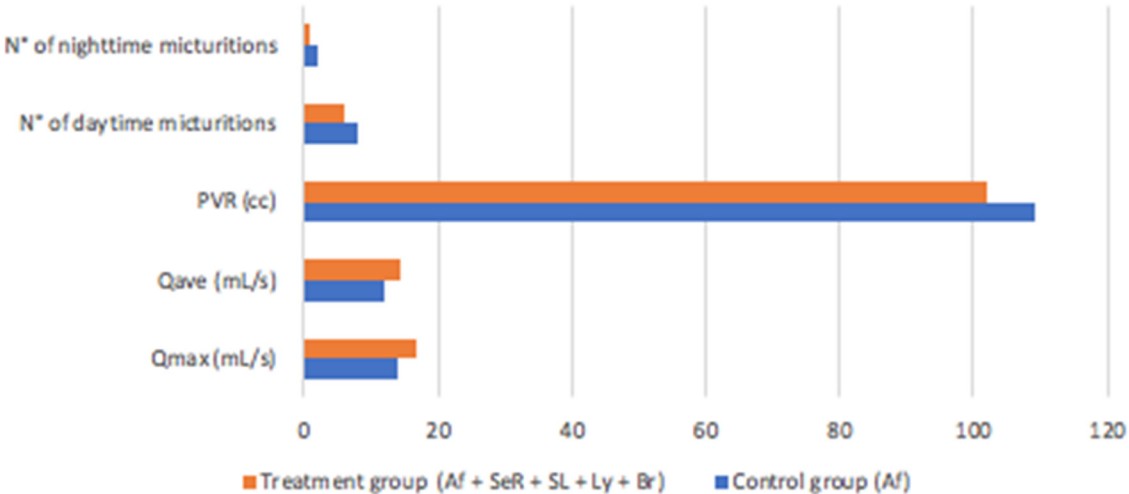

**Figure 4.** Descriptive statistics of the efficacy parameters at the 6-month follow-up assessment.

## 4. Discussion

This multicenter, prospective study was designed to assess clinical and functional outcomes in men presenting with mild to moderate LUTS previously managed with Alfuzosin 10 mg monotherapy during a 6–12 months period and treated with a combination therapy of Alfuzosin and SeR + SL + Ly + Br for 6 months in a real-life setting. A retrospective control group was obtained on patients previously managed with Alfuzosin 10 mg/day during a 6–12 months between March 2015 and December 2018 by performing a propensity score–matched analysis. Of note, patients treated with Alfuzosin and SeR + SL + Ly + Br reported significant improvement in terms of IPSS and its storage and voiding subscores as well as QoL compared to the Alfuzosin monotherapy control group.

Over the last decades, phytotherapeutic and nutraceutical agents gained widespread interest in the scientific panorama as monotherapy and/or combination with alpha-blockers in patients with BPH-related LUTS [16]. Nevertheless, current international guidelines do not specifically recommend their use in everyday clinical practice and several questions still remain unanswered [2]. Indeed, trustworthiness of current available literature is undermined by the heterogeneity of patients enrolled, the lack of adequately long-term follow-up and the use of unvalidated symptom-scale scores. In addition, the different kinds of extraction still limit the correct interpretation of the available data [17]. Currently, SeR represents the most commonly used phytotherapeutic compound for BPH management [18] and the only one actually recommended for well-established use by the Committee on Herbal Medicinal Products (HMPC). Its effectiveness in prostate size reduction and urinary frequency improvement was also confirmed by two systematic reviews and meta-analyses [10,19]. Conversely, SeR showed not to influence storage and voiding LUTS as well as PSA serum levels [20–22]. However, while most studies mainly focused on nutraceutical agent effects as monotherapy, only few studies have investigated the benefit of combination therapy with AB reporting unequivocal results.

Recently, Morgia et al. showed that combination treatment of SeR-Se-Ly with tamsulosin was more effective than monotherapy in improving the IPSS and Qmax after one-year follow-up (11). In another study, Ryu et al. [23] reported that the combination of SeR and tamsulosin was more effective than tamsulosin as monotherapy in reducing storage symptoms. Conversely, Hizli et al. [24] evaluated the efficacy of SeR plus tamsulosin vs. both SeR and tamsulosin as monotherapy, reporting no differences among groups in terms of IPSS and Qmax. Similarly, the TAMSR trial [25] proved that combination therapy did not lead

to any meaningful clinical advantage regarding IPSS, Qmax, or PVR. Nevertheless, these studies were not devoid of several limitations, including the lack of detailed information about the supplementation interruption assessment and the kind of extraction performed which may have eventually influenced the quality of the compound.

In this light, the present study was designed comparing a matched-paired cohort of 125 patients managed with Alfuzosin 10 mg monotherapy with 125 consecutive patients with mild to moderate LUTS previously managed only with Alfuzosin 10 mg and treated with a combination treatment of Alfuzosin plus SeR + SL + Ly + Br (Perlaprost®) over a 6 month period. We found that patients in the treatment group had a significantly improvement of the median IPSS score after 6 months of supplementation with similar results in terms of both storage and voiding symptoms. Particularly, a significative improvement was recorded in both storage subscore and in median number of night-time micturition. Conversely, unlike previous studies [11,24,26], Qmax significantly improved after the 6-month supplementation comparing to the control group. Similarly, a significative decrease of PVR was reported at each follow-up visit, while only few studies in current literature reported SeR meaningfully influencing PVR [2,16,22]. These findings may be explained by the higher concentration of supercritical $CO_2$ lipidic extract contained in oral tablets of Perlaprost® which might maximize its anti-inflammatory and anti-edematous activity. Indeed, several series suggested SeR inhibition of cyclooxygenase and 5-lipoxygenase metabolites production and reduction in monocyte chemoattractant protein-1/Chemokine (C-C) motif ligand 2 (MCP-1/CCL/2) which stimulates monocyte recruitment and activation during inflammation [27]. Moreover, also the implementation of bromelain with an high concentration may act particularly on storage symptoms relief by enhancing the production of prostaglandins with anti-inflammatory activity [28,29]. Most importantly, in our experience, total PSA was significantly decreased after 3 months of combination treatment with no persisting benefit following the 6-month supplementation interruption. In this regard, the presence of both phytotherapeutic and nutraceutical extracts as lycopene and Solanum lycopersicum may have played a pivotal role. Indeed, the anti-inflammatory and anti-proliferative effects of these agents have been previously reported by Ford et al. [30] who found a significative reduction of oxidative DNA damage in prostatic tissue and a local androgen signaling downregulation in patients with dietary supplementation of SL + Ly. In this light, all these properties are likely to counteract chronic prostatic inflammation playing a role in the relief of BPH-related LUTS.

To the best of our knowledge, this represents the first study investigating clinical and functional outcomes of combination treatment with Alfuzosin and SeR + SL + Ly + Br in a real-life setting. Major strengths of the present study were the strict inclusion criteria, defined dosage, and extraction of phytotherapeutic and nutraceutical agents and use of validated symptom-scale scores during a mid-term follow-up period.

The current study was not devoid of several limitations. First, we acknowledge the relatively low sample size and the retrospective nature of the study. Second, although a matched-paired analysis was performed to lower study population inhomogeneity, the lack of a prospective control arm might have introduced non-negligible statistical bias. Moreover, the lack of assessment of clinical and functional outcomes within a study population managed with different kinds of alpha-blockers should limit the strength of the clinical message provided.

## 5. Conclusions

Acknowledging the aforementioned limitations, our findings suggest the efficacy and safety of the combination of alpha-blocker plus SeR + SL + Ly + Br for the management of symptomatic BPH leading to a meaningful improvement in LUTS severity compared to AB as monotherapy, after a 6-month treatment period in men with mild to moderate LUTS/BPH. However, we are aware that larger, randomized studies with longer follow up are warranted to confirm our preliminary findings.

**Author Contributions:** Research conception and design: A.M. (Andrea Mari), L.L., R.T., L.M., A.M. (Andrea Minervini); Data acquisition: F.V., C.B.; Statistical analysis: A.M. (Andrea Mari), L.L., R.T., A.A.G.; Data analysis and interpretation: L.L., A.M. (Andrea Mari), S.S.; Drafting of the manuscript: F.D.M., L.L., M.G.; Critical revision of the manuscript: G.M.P., G.M.B., M.F., F.D.G., O.d.C.; Administrative, technical, or material support: L.M., A.M. (Andrea Minervini); Supervision: A.M. (Andrea Mari), L.M., A.M. (Andrea Minervini); Approval of the final manuscript: A.M. (Andrea Mari), A.M. (Andrea Minervini). All authors agree to be accountable for all aspects of the work in ensuring that questions related to the accuracy or integrity of any part of the work are appropriately investigated and resolved. All authors have read and agreed to the published version of the manuscript.

**Funding:** This research received no external funding.

**Institutional Review Board Statement:** The study was conducted according to the guidelines of the Declaration of Helsinki, and approved by the Institutional Review.

**Informed Consent Statement:** Informed consent was obtained from all subjects involved in the study.

**Data Availability Statement:** Reported data are in possession of corresponding author fully available if requested.

**Conflicts of Interest:** The authors certify that there are no conflict of interest with any financial organization regarding the material discussed in the manuscript.

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
