# Peer review of "Impact of the Treatment of Serenoa repens, Solanum lycopersicum, Lycopene and Bromelain in Combination with Alfuzosin for Benign Prostatic Hyperplasia. Results from a Match-Paired Comparison Analysis"

_2673-4397, doi:10.3390/uro1040025_

Round 1

Reviewer 1 Report

The manuscript is quite interesting but there are some aspects that need to be improved: first of all the study design is not very clear - it is not easy to understand if the study group received Alfuzosin. Also, the Conclusion is not mentionned after the Discussion section, even if it is mentionned in the abstract. 

Author Response

  1. The manuscript is quite interesting but there are some aspects that need to be improved: first of all the study design is not very clear - it is not easy to understand if the study group received Alfuzosin.

We kindly thank the reviewer for his accurate revision and the overall appreciation of the manuscript. Patients enrolled within the treatment group were managed with a daily combination treatment of Alfuzosin 10mg + Serenoa Repens + Solanum Lycopersicum + Lycopene + Bromelain (SeR + SL + Ly + Br). The study design has been represented in a dedicated flow chart order to better describe the methods.  

  1. Also, the Conclusion is not mentionned after the Discussion section, even if it is mentionned in the abstract. 

We really appreciate reviewer’s suggestion, we provided to integrate a Conclusion section also within the manuscript.

Reviewer 2 Report

I congratulate with the authors for this well written multicentric paper comparing treatment with phythotherapics + alfuzosin with conventional alfuzosin only. This showed improved symptoms in the treatment group.

I understand that the patients enrolled to the mix of phytotherapies were already on Alfuzosin treatment and this was not stopped during the study. However, reading your inclusion criteria looks like they have to discontinue Alfuzosin to be able to be enrolled. Please change :  "previous therapy with 
 Alfuzosin 10 mg over a 6-12-month" period  with "Already on treatment with Alfuzosin 10 mg for at least 6-12-month".

I think the comparison was tied to Alfuzosin because this drug was used in the retrospective cohort, but I think that using Tamsulosin or Silodosin might have led to better results in the control arm too.

Why did the author not perform a randomised controlled trial rather than comparing with retrospective cohort?

regards

Author Response

1. I understand that the patients enrolled to the mix of phytotherapies were already on Alfuzosin treatment and this was not stopped during the study. However, reading your inclusion criteria looks like they have to discontinue Alfuzosin to be able to be enrolled. Please change :  "previous therapy with 
 Alfuzosin 10 mg over a 6-12-month" period  with "Already on treatment with Alfuzosin 10 mg for at least 6-12-month".

We appreciate reviewers suggestion, in order to better clarify the study design we provided to correct "previous therapy with 
 Alfuzosin 10 mg over a 6-12-month" with "Already on treatment with Alfuzosin 10 mg for at least 6-12-month". Moreover, we integrated the study design flow chart in a dedicated figure. 

I think the comparison was tied to Alfuzosin because this drug was used in the retrospective cohort, but I think that using Tamsulosin or Silodosin might have led to better results in the control arm too.

Why did the author not perform a randomised controlled trial rather than comparing with retrospective cohort?

We kindly thank the reviewer for outlining this crucial issue of paramount importance. As outlined within the ''Materials and Methods'' sections, the inclusion criteria incorporates 

1) age between 40 and 70 years 2) previous therapy with Alfuzosin 10 mg over a 6-12-month period 3) International prostate symptom score (IPSS) <20 with a positive answer to urgency and frequency questions.   

In this light, Alfuzosin represents an effective first-line treatment particularly in patients with mild to moderate LUTS (IPSS < 20) due to the lower rate of ejaculatory disorders without compromising the overall Blood Pressure control.    

We selected very strict inclusion criteria to create two groups of patients with similar baseline conditions.
Indeed, this is only a retrospective study and definitive conclusions cannot be drawn with this type of evidence. Nevertheless, this study showed promising findings reporting a significant reduction of storage symptoms in selected patients with BPH treated with Perlaprost in addition to alfuzosin. This will be a crucial point to design prospective trials to test the efficacy of combinations of different phytotherapeutics in addition to alpha-blockers to improve QoL and reduce storage symptoms in selected patients with BPH.

Nevertheless, we integrated these limitations within the discussion section.  

Round 2

Reviewer 1 Report

The manuscript could be published in this form.